# Kinetics of Dehydroxylation and Decarburization of Coal Series Kaolinite during Calcination: A Novel Kinetic Method Based on Gaseous Products

**DOI:** 10.3390/ma14061493

**Published:** 2021-03-18

**Authors:** Simeng Cheng, Shaowu Jiu, Hui Li

**Affiliations:** 1College of Materials Science and Engineering, Xi’an University of Architecture and Technology, Xi’an 710055, China; chengsimeng@xauat.edu.cn (S.C.); jiushaowu@xauat.edu.cn (S.J.); 2Shaanxi Ecological Cement Concrete Engineering Technology Center, Xi’an 710055, China

**Keywords:** coal series kaolin, TG-FTIR-MS, kinetics, reaction mechanism

## Abstract

The analysis of gaseous products reveals the characteristics, mechanisms, and kinetic equations describing the dehydroxylation and decarburization in coal series kaolinite. The results show that the dehydroxylation of coal series kaolinite arises from the calcination of kaolinite and boehmite within the temperature range of 350–850 °C. The activation energy for dehydroxylation is 182.71 kJ·mol^−1^, and the mechanism conforms to the A2/3 model. Decarburization is a two-step reaction, occurring as a result of the combustion of carbon and the decomposition of a small amount of calcite. The temperature range in the first step is 350–550 °C, and in the second is 580–830 °C. The first step decarburization reaction conforms to the A2/3 mechanism function, and the activation energy is 160.94 kJ·mol^−1^. The second step decarburization reaction follows the B3 mechanism function, wherein the activation energy is 215.47 kJ·mol^−1^. A comparison with the traditional methods proves that the kinetics method utilizing TG-FTIR-MS is feasible.

## 1. Introduction

Coal plays a vital role in the current energy supply worldwide, especially in China [1]. Coal series kaolinite (CSK) is one of the foremost types of coal gangue, with confirmed reserves of 1.673 billion tons and prospective reserves of 5.529 billion tons in China alone [2]. Worldwide, CSK is also distributed in major kaolin producing countries such as the United States, Australia, the United Kingdom, Russia, the Czech Republic, Germany, and Brazil, among others. CSK is typically abandoned as coal mine waste, resulting in serious ecological and environmental pollution [3,4]. However, CSK is an ideal resource for replacing high-quality natural kaolin. Calcination is a typical method to achieve high chemical reactivity of CSK prior to further applications, such as preparation of calcined kaolin products and metakaolin cementitious materials [5,6]. Dehydroxylation and decarburization are the main reactions during the calcination of CSK [7,8,9]. The activity of metakaolin prepared from CSK depends essentially on the dehydroxylation reaction. However, the whiteness of addition agents from CSK (used in paints, plastics, coatings, and ceramics) is significantly affected by the decarburization reaction. Therefore, it is essential to characterize the dehydroxylation and decarburization reactions in CSK and to study their reaction characteristics, mechanism, and kinetics.

The traditional thermal analysis method, based on the data of mass or heat by Thermogravimetric analysis-Differential Scanning Calorimetry (TG-DSC), is commonly used for reaction kinetics [10,11,12,13,14,15]. The decarburization and dehydroxylation reactions in CSK are synchronous, and the mass loss and exothermic heat generation are almost simultaneous. Therefore, the decarburization and the dehydroxylation in CSK cannot be characterized by either the mass or heat, which is a major challenge for CSK calcination. The kinetics calculated from TG/DSC data present the apparent characteristics of the whole calcination reaction of CSK, but not the real characteristics of dehydroxylation and decarburization [5,16,17]. Nonetheless, the gaseous products released by the reactions in CSK are different, which can be distinguished by detecting product release. With the development of thermal analysis technology, Thermogravimetric analysis-Fourier Transform Infrared Spectrometer-Mass Spectrometer (TG-FTIR-MS) technology has been widely used to detect gaseous products [18,19,20]. In TG-FTIR-MS, the gaseous products can be detected and identified by the FTIR detector. The release flows of gaseous products can be recorded by MS, providing an effective means for analyzing complex reactions [21]. There are several applications of TG-FTIR and TG-MS, combined for the analysis of reaction mechanisms and kinetics, but there has been no report till date on the kinetics of CSK [22,23,24]. Hence, it is necessary to study the mechanisms and kinetics involved in the dehydroxylation and decarburization of CSK based on the analysis of gaseous products by TG-FTIR-MS.

In the present work, the TG-FTIR-MS method was used to identify and quantify the gaseous products of CSK under calcination. Three kinetic methods were used by gas product flow data instead of TG/DSC to calculate the kinetic equations. The mechanisms and kinetics of dehydroxylation and decarburization during the calcination of CSK were obtained, respectively. A comparison with the traditional methods revealed the advantages of the kinetic method by TG-FTIR-MS. The work in this paper provided a theoretical basis for the fine control of dehydroxylation and decarburization in CSK calcination. In addition, it provided guidance for the process design, energy saving, and pollutant control of green building materials and high-quality calcined kaolin products prepared by CSK.

## 2. Experiment and Methods

### 2.1. Raw Materials

The CSK samples were taken from interbedded coal gangue of Junggar coalfield in Inner Mongolia, Ordos, China. A total of 30 tons of samples collected in the same batch were broken into small pieces of about 3–5 cm by a crusher. Approximately 50 kg of CSK was taken out by multi-point sampling method and broken into particles with a particle size of less than 3 mm. A ceramic ball mill was used to grind the sample into powder with a particle size of less than 80 μm. Before grinding, the powder sample was dried in an electric oven at 105 °C for 4 h. The raw materials were characterized by X-ray diffractometer (XRD), X-ray fluorescence (XRF), and flammability analysis.

#### 2.1.1. XRD Analysis

An X-ray diffractometer (XRD, D/MAX-2200, Japan Rigaku Corporation, Akishima City, Tokyo, Japan) was used to analyze the mineral composition of the raw material. The conditions of the instrument were Cu target Kα ray, tube voltage of 45 kV, and tube current of 40 mA. Figure 1 shows the XRD patterns of the CSK.

The CSK sample largely contained kaolinite, boehmite (AlO(OH)), and calcite (CaCO_3_), as well as a small amount of oxidized minerals of Pb, Zr, and Ti.

#### 2.1.2. XRF Analysis

An X-ray fluorescence spectrometer (XRF, S4PIONEER, German Bruker Company, Karlsruhe, Germany) was used for elemental analysis of the raw material. The X-ray tube parameters were 4.2 kW, 60 kV (Max), and 140 mA (Max). Table 1 provides the results of the XRF analysis of CSK.

The primary chemical elements present in the CSK were silicon, aluminum, calcium, and titanium, along with some trace elements, such as iron, potassium, magnesium, sulfur, and phosphorus. According to the semi-quantitative calculation from Figure 1 and Table 1, the contents of kaolinite and boehmite in CSK are 72.82 wt.% and 19.17 wt.%, respectively.

#### 2.1.3. Flammability Analysis

The flammability of CSK was analyzed by the method specified in the Chinese Coal Industry Analysis Standard (GB/T212-2008). The analysis results are shown in Table 2. The carbon content of CSK is 2.62 wt.%.

### 2.2. TG-FTIR-MS Analysis

A thermal analyzer (NETZSCH 409PC STA, German), infrared spectrometer (Bruker FTIR-7600, Karlsruhe, Germany), and mass spectrometer (Perkin Elemer SQ8T, Norwalk, CT, USA) were used to study the calcination of CSK. The ion source of MS was an electron bombardment source, and its working temperature was 250 °C. The temperature of the transmission pipeline was 280 °C, the electron bombardment energy was 780 eV, and the scanning mode was the full scanning mode. The test parameters were 90% N_2_ + 10%O_2_ atmosphere (similar to the composition of industrial waste gas); flow rate of 75 mL/min; heating rates of 5 °C/min, 10 °C/min, 15 °C/min, and 20 °C/min; sample mass of 6.0 ± 0.5 mg; and temperature range from ambient to 1000 °C. By analyzing the TG, FTIR, and MS data, the gaseous products released in the process of CSK calcination were identified and quantified. The characteristics for dehydroxylation and decarburization of CSK were analyzed by analyzing the mass spectrum data of gas products.

### 2.3. Theory and Mathematical Methods

A typical kinetics equation for the solid-state reduction process can be expressed as follows [25,26,27]:dα/dt = k(T)f(α)(1)
where α is the degree of decomposition, dα/dt is the decomposition rate, and f(α) is the function of α. k(T) pertains to the temperature-dependent decomposition rate constant, expressed by the Arrhenius equation as follows:k(T) = A·exp(−E_a_/RT)(2)
where A is the pre-exponential factor (s^−1^), R denotes the gas constant (8.314 J·K^−1^mol^−1^), and T is the absolute temperature (K). E_a_ represents the apparent activation energy (kJ·mol^−1^), typically termed as the minimum energy required to overcome the energy barrier for the reaction and to form products [28].

For the linear heating rate test method, the heating rate is constant. The relationship between β, T, and t is as follows:β = dT/dt(3)

Substituting Equations (2) and (3) into Equation (1), the expression of reaction rate is obtained as follows:dα/dT = A/β·exp(−E_a_/RT)f(α)(4)

Equation (5) can be obtained by separating the variables and rearranging and integrating Equation (4).
 G(α) = ∫^0^αdα/f(α) = A/β·∫_T0_^T^exp(−E_a_/RT)dT(5)
where G(α) is the integral form of the reaction mechanism function and T_0_ is the initial temperature of the non-isothermal experiment. Equation (5) is the kinetic equation in the integral calculus form.

The degree of decomposition (conversion) α characterized by TG-FTIR-MS is as follows:α = S_t_/S_∞_(6)
where S_t_ and S_∞_ are the partial and total areas of the peak in the mass spectrum curve of gaseous products, respectively.

The general integral method, the Mac Callum–Tanner method, and the Satava–Sestak method were used to calculate the mechanism function G(α) and pre-exponential factor A [29].

The equation for the general integration method is as follows:ln[G(α)/T^2^] = ln{(AR/βE_a_)[1 − 2(RT/E_a_)]} − E_a_/RT(7)

The equation of the Mac Callum–Tanner method is as follows:ln[G(α)] = ln{(AE_a_/βR) − 0.4828E_a_^0.4357^ − (449 + 217E_a_)/T(8)

The Satava–Sestak equation is presented as follows:lgG(α) = lg(AE_a_/Rβ) − 2.315 − 0.4567E_a_/RT(9)

In accordance with the linear relationship, the optimal mechanism function can be obtained by the least square method. Apparent activation energy E_a_ and pre-exponential factor A were calculated from the slope and intercept of the fitted line, respectively. Table 3 lists quite a few of the common mechanism functions of G(α) for solid-state reactions [29].

The self-developed kinetics program based on these theories was used to calculate the kinetics of dehydroxylation and decarburization of CSK.

## 3. Results and Discussion

### 3.1. Thermal Analysis

The TG–DTG curves of CSK are shown in Figure 2.

The weight loss of CSK during heating largely occurred within the range of 330–850 °C. A small and hardly visible mass loss observed in the temperature range of 50–150 °C (DTG, shown in Figure 2) is associated with the loss of adsorbed water [30]. There were two stages of CSK calcination distinctly visible from DTG. The first stage started at 330 °C and ended at 590 °C–640 °C. The second stage lasted until about 850 °C. At the end of the second stage, the weight of the sample hardly underwent any change. The characteristic temperatures are listed in Table 4. The mass loss at different heating rates is shown in Figure 3.

The mass losses in the first stage were 13.71 wt.%, 13.63 wt.%, 11.95 wt.%, and 11.89 wt.%, corresponding to the heating rates of 5 °C/min, 10 °C/min, 15 °C/min, and 20 °C/min, respectively. The average value of the first stage at these four heating rates is 12.80 wt. %. The second stage mass losses were 5.33 wt.%, 4.53 wt.%, 4.22 wt.%, and 3.58 wt.%, with an average value of 4.42 wt.%. The first stage mass loss is about three times that of the second stage, indicating that the first stage reaction is the dominant one. As the heating rate increases, the mass loss decreases, which is similar to the phenomenon reported in the literature [30]. This phenomenon can be explained from the perspective of heat transfer. With the increase in heating rate, the time needed to reach a specific temperature is shortened. Thus, the material with larger particles is not evenly heated, resulting in the reduction of the degree of reaction [31,32].

### 3.2. Infrared Analysis

Figure 4 presents the infrared absorption spectrum (3D) of gaseous products released in the thermal analysis at the 10 °C/min heating rate. The results of infrared spectrum analysis of gaseous products at different times are shown in Figure 5.

In Figure 5, the infrared absorption at 3700 cm^−1^ is associated with the inner-surface O-H stretching, and that at position 3660 cm^−1^ is attributed to the inner-cage O-H stretching of kaolinite [33]. The hydroxyl groups bending vibrations exist at 1518 cm^−1^ and 1750 cm^−1^ [34]. Asymmetrical stretching and bending vibrations of CO_2_ occur at 2360 cm^−1^ [35]. Thus, the infrared absorption of gaseous products released during the CSK calcination is mainly due to O-H and C-O groups. As the major mineral phases in the sample are kaolinite, boehmite, and calcite, it can be concluded that the gaseous products of CSK calcination are H_2_O_(g)_ and CO_2_. Comparing Figure 4 with Figure 5, peak 1 and peak 3 in Figure 4 correspond to the absorption of H_2_O_(g)_, and peak 2 corresponds to the absorption of CO_2_.

### 3.3. Mass Spectrometry Analysis

In a mass spectrometer, H_2_O_(g)_ and CO_2_ in gaseous products produce various fragments with different charge-to-mass ratios. The charge-to-mass ratios of M_18_ and M_44_ were selected to represent H_2_O_(g)_ and CO_2_, respectively. The release intensities (abundance) at different temperatures of M_18_ and M_44_ by MS are shown in Figure 6 and Figure 7, respectively.

As shown in Figure 6, the release of H_2_O_(g)_ is largely from 350 and 750 °C, which indicates that the dehydroxylation of CSK occurs in that temperature range. There is a single peak in the release curves of H_2_O_(g)_ at different heating rates, indicating that the dehydroxylation of CSK is a continuous reaction. The temperatures for the maximum dehydroxylation rate were 518.3 °C, 536.7 °C, 554.3 °C, and 564.5 °C at the heating rates of 5 °C/min, 10 °C/min, 15 °C/min, and 20 °C/min, respectively. The peak of the release curve moves to the high-temperature region with the increase in heating rate, consistent with the trend in Figure 2.

There are two peaks in the release intensity curve of M_44_ at different heating rates (as shown in Figure 7), indicating that the decarburization of CSK was a two-step reaction. The temperature range of the first step decarburization was roughly 350–620 °C, and that of the second was 580–840 °C. The carbon found in CSK samples mainly comes from coal and a small amount of calcite. As the decomposition temperature of calcite is higher than 800 °C, the two-step reaction displayed the actual characteristics of CSK decarburization [36]. The peak area on the right side of the curves in Figure 7 was much larger than that on the left side, indicating that the second step reaction was the major CSK decarburization stage.

Figure 8 shows the temperature ranges of the CSK calcination stage by TG/DTG and that by MS at four heating rates.

As shown in Figure 8a–d, there is a significant overlap between the temperature ranges for the dehydroxylation and decarburization during CSK calcination. The temperature range for decarburization covered the whole temperature range of dehydroxylation. Nonetheless, the two reaction stages divided by TG/DSC did not conform to either dehydroxylation or decarburization, thus the physical meaning was not clear. Hence, the kinetics calculated according to the temperature range by TG/DSC did not provide an actual physical meaning.

### 3.4. Kinetics Analysis

The kinetics of dehydroxylation and decarburization in CSK calcination were analyzed from MS data. The conversion curve for dehydroxylation of CSK is shown in Figure 9, calculated from MS curves in Figure 6 by Equation (6).

The decarburization of CSK was a two-step reaction. Hence, the conversion curves for each step reaction were calculated separately. The conversion rate curves of the first step decarburization and the second decarburization are shown in Figure 10a,b, calculated from the integration of the left and the right peaks in Figure 7, respectively.

The results for the dehydroxylation as well as the first step and second step in the decarburization of CSK are listed in Table 5, Table 6 and Table 7, respectively.

The absolute values of the maximum deviation of activation energy listed in Table 5, Table 6 and Table 7 are 6.37%, 4.45%, and 5.40%. The linear correlation coefficients are greater than 0.989, which indicates the kinetics results are reliable. The physical significance and mechanism equations of No. 17 (in Table 5 and Table 6), No. 29 (in Table 7), and No. 37 are listed in Table 8.

The dehydroxylation reaction in CSK calcination conforms to the A2/3 mechanism model, wherein the activation energy is 182.71 kJ·mol^−1^. Nonetheless, the mechanism function and activation energy in the two-stage decarburization reaction in CSK are different. The first step decarburization reaction also fits the A2/3 mechanism model, wherein the activation energy is 160.94 kJ·mol^−1^. The second step decarburization reaction fits into the B3 mechanism model, with the activation energy being 215.47 kJ·mol^−1^. The activation energy of the second decarburization step is greater than that of the dehydroxylation reaction, indicating that the decarburization is more difficult than dehydroxylation in CSK calcination. Therefore, the kinetics equations of the dehydroxylation and decarburization reactions in CSK calcination are as follows:[−ln(1−α)]^3/2^ = 10^9.44^exp(−182.71/RT)t (dehydroxylation reaction)(10)
[−ln(1−α)]^3/2^ = 10^9.02^exp(−160.94/RT)t (the first step of decarburization)(11)
1−(1−α)^1/3^ = 10^9.37^exp(−215.47/RT)t (the second step of decarburization)(12)

### 3.5. Comparative Analysis of New Methods and Traditional Methods

The kinetics results obtained from TG are shown in Table 9 and Table 10, and the comparison with the new kinetic method (TG-FTIR-MS) is shown in Table 11.

The activation energy is 154.11 kJ·mol^−1^ and the mechanism function is No. 17 of CSK calcination at the first stage of mass loss, as obtained from TG method, similar to the literature [12,14]. The first stage mass loss is mostly caused by the dehydroxylation reaction, thus the reaction mechanism obtained by TG is the same as that obtained from TG-FTIR-MS. The second stage mass loss of TG is mainly the superposition in the later period of the dehydroxylation and the second step of decarburization (shown in Figure 8). The activation energy and the mechanism in the second stage mass loss of TG are different from that of the dehydroxylation reaction and the second step of the decarburization reaction. Therefore, it can be inferred that the kinetics obtained by the TG method may not have physical meaning when the mass loss between the reactions is not much different. The TG-FTIR-MS method can be used to characterize the dehydroxylation and decarburization characteristics and kinetics of CSK in more detail, which is more reasonable than the traditional TG method. The new kinetic method revealed the real kinetic characteristics of the dehydroxylation and decarburization in CSK, which provided a theoretical basis for the calcination control of CSK. In practical applications, the new method can easily reveal the differences in the reaction characteristics of different types of CSK. The obtained kinetic equations can be used to predict the reaction curves of dehydroxylation and decarburization in CSK under various calcination conditions. All these have guiding significance for process design, product quality control, energy saving, and pollutant control of CSK calcination.

## 4. Conclusions

(1)The dehydroxylation reaction in CSK calcination conforms to the A2/3 mechanism model, the temperature range is 350–850 °C, and the activation energy is 182.71 kJ·mol^−1^. The decarburization in CSK calcination is a two-step reaction; the temperature range in the first step is 350–550 °C, and in the second is 580–830 °C. The first step decarburization reaction conforms to the A2/3 mechanism function, and the activation energy is 160.94 kJ·mol^−1^. The second step decarburization reaction follows the B3 mechanism function, wherein the activation energy is 215.47 kJ·mol^−1^.(2)The TG-FTIR-MS method is suitable for analyzing the characteristics and kinetics of dehydroxylation and decarburization in CSK calcination. It is also a theoretical tool for the kinetic analysis of other types of complex reactions.

## Figures and Tables

**Figure 1 materials-14-01493-f001:**
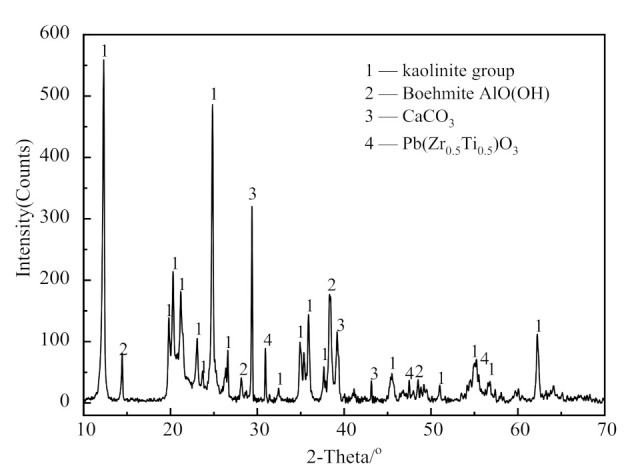
X-ray diffractometer (XRD) patterns of coal series kaolinite (CSK).

**Figure 2 materials-14-01493-f002:**
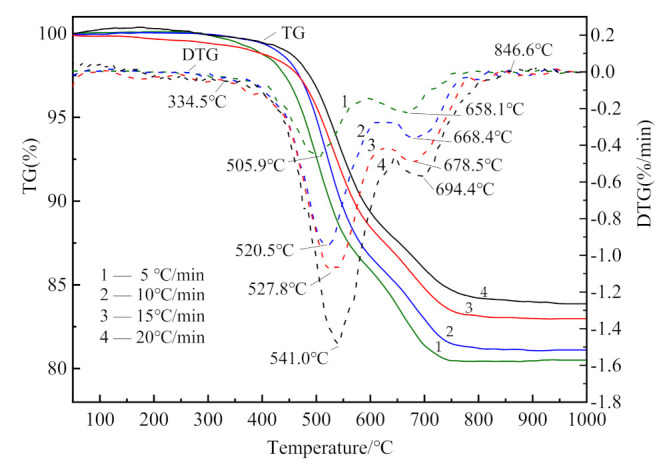
TG and DTG curves of CSK at different heating rates.

**Figure 3 materials-14-01493-f003:**
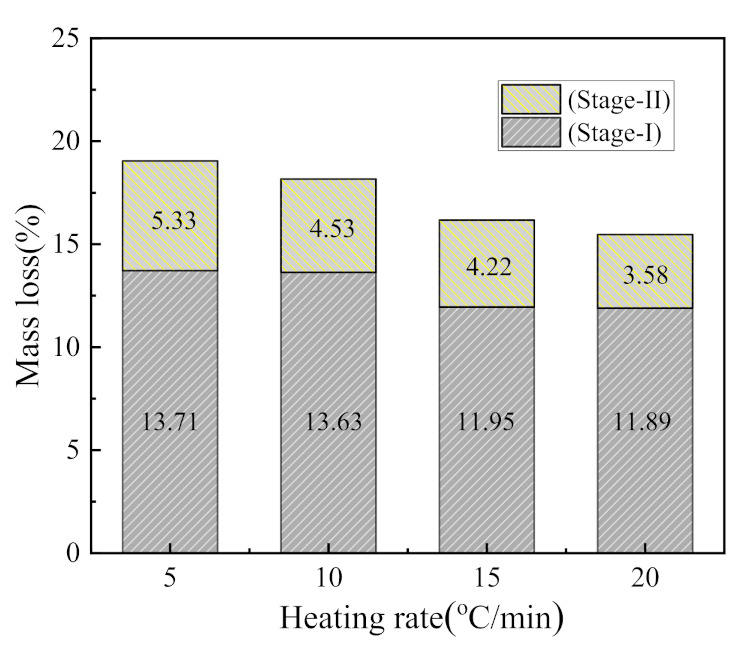
The mass loss during CSK calcination at different heating rates.

**Figure 4 materials-14-01493-f004:**
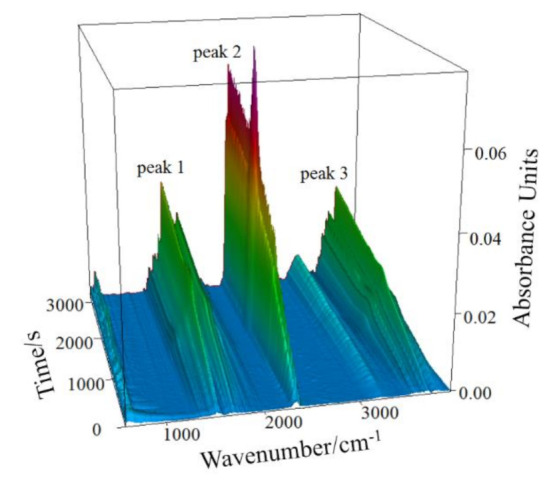
Infrared absorption spectrum (3D) of gaseous products from CSK at the heating rate of 10 °C/min.

**Figure 5 materials-14-01493-f005:**
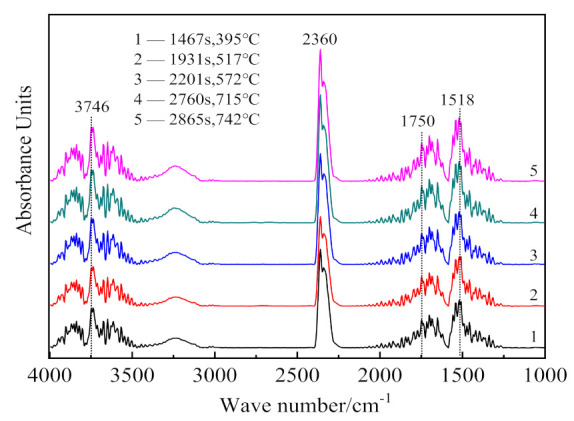
Analysis results of infrared absorption spectra of gaseous products at different times (temperature).

**Figure 6 materials-14-01493-f006:**
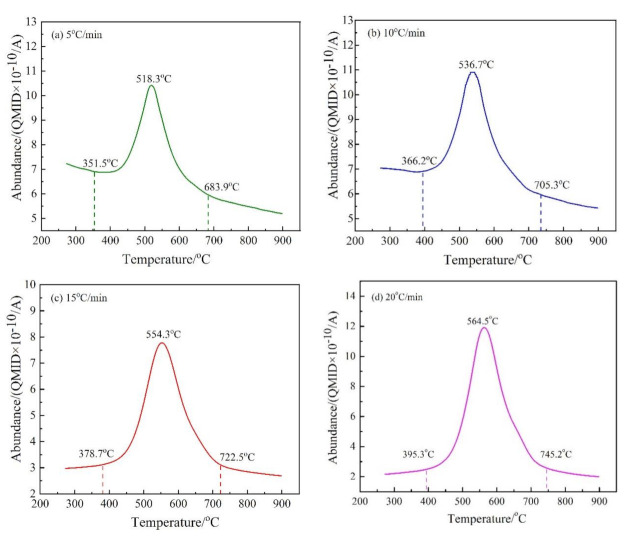
The release intensity (abundance) curves of M/z = 18 at different heating rates. (**a**) 5 °C/min; (**b**) 10 °C/min; (**c**) 15 °C/min; and (**d**) 20 °C/min.

**Figure 7 materials-14-01493-f007:**
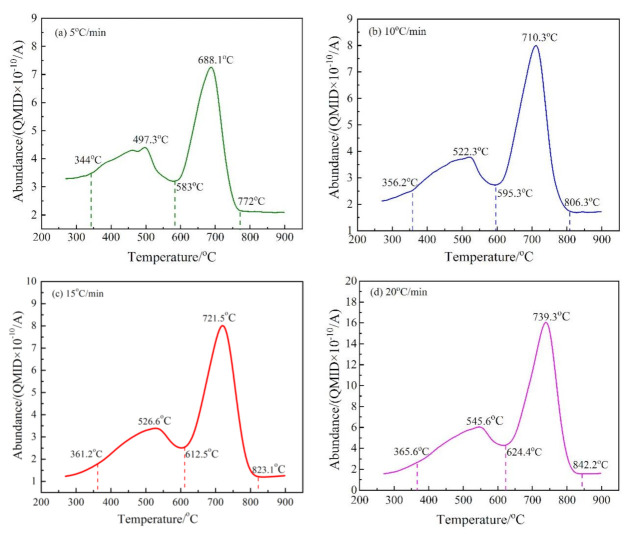
The release intensity (abundance) curves of M/z = 44 at different heating rates. (**a**) 5 °C/min; (**b**) 10 °C/min; (**c**) 15 °C/min; and (**d**) 20 °C/min. *QMID in the figure is the abbreviation of “quasi multi-ion detection”.

**Figure 8 materials-14-01493-f008:**
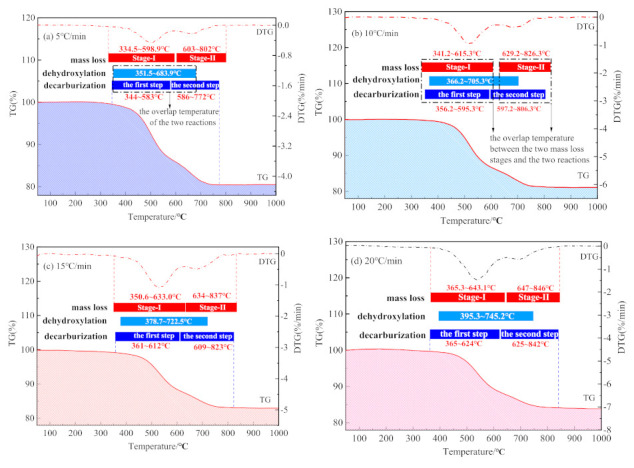
Comparison of reaction temperature range determined by TG/DSC and MS. (**a**) 5 °C/min; (**b**) 10 °C/min; (**c**) 15 °C/min; and (**d**) 20 °C/min.

**Figure 9 materials-14-01493-f009:**
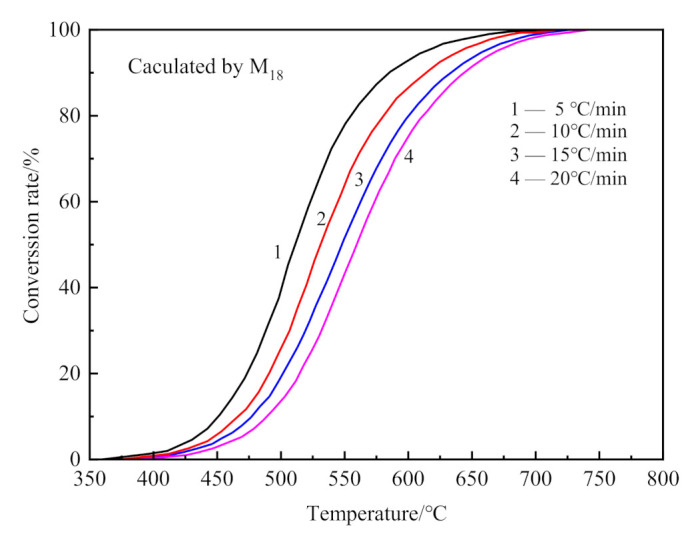
The conversion curves for dehydroxylation of CSK at different heating rates.

**Figure 10 materials-14-01493-f010:**
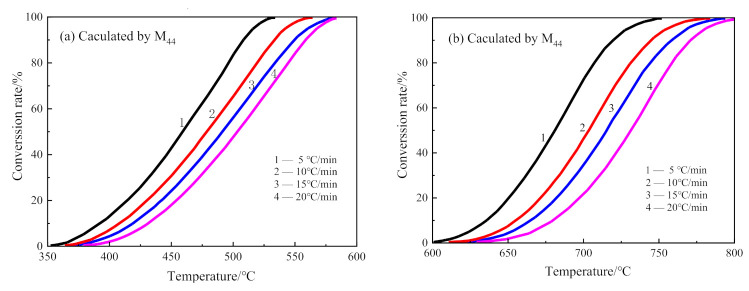
The conversion curves for decarburization of CSK at different heating rates. (**a**) The first step of CSK decarburization; (**b**) the second step of CSK decarburization.

**Table 1 materials-14-01493-t001:** Chemical composition of raw material (wt.%).

SiO_2_	Al_2_O_3_	CaO	TiO_2_	Fe_2_O_3_	K_2_O	MgO	SO_3_	P_2_O_5_	PbO	Zr_2_O_3_
38.42	34.77	4.20	1.35	0.39	0.13	0.08	0.05	0.07	0.67	0.18

**Table 2 materials-14-01493-t002:** Industrial analysis results of coal series kaolin samples (wt.%).

Moisture	Ash	Volatile	Total Sulfur	Total Moisture	Carbon	Hydrogen	Nitrogen
1.00	82.8	14.74	0.03	1.06	2.62	1.53	0.07

**Table 3 materials-14-01493-t003:** Integral mechanism functions used for solid-state reactions.

Symbol of g(α)	The Sequence Number of Function	Equation Name	Expression of G(α) Function
D1	1	One-dimensional diffusion	*α* ^2^
D2	2	Two-dimensional diffusion	*α +* (1 − *α*)ln(1 − *α*)
1D3	6	Tri-dimensional diffusion (spherically symmetric)	[1 − (1 − *α*)^1/3^]^2^
2D3	7	Tri-dimensional diffusion (cylindrically symmetric)	1 − 2*α*/3 − (1 − *α*)^2/3^
A1	16	Nucleation and growth (*n* = 1)	−ln(1 − *α*)
A2/3	17	Nucleation and growth (*n* = 3/2)	[−ln(1 − *α*)]^3/2^
A2	13	Nucleation and growth (*n* = 1/2)	[−ln(1 − *α*)]^1/2^
A3	11	Nucleation and growth (*n* = 1/3)	[−ln(1 − *α*)]^1/3^
R2	31	Shrinking core (cylindrically symmetric)	1 − (1 − *α*)^1/2^
R3	29	Shrinking core (spherically symmetric)	1 − (1 − *α*)^1/3^
P2	24	Power law	*α* ^1/2^
P3	23	Power law	*α* ^1/3^
C2	37	Chemical reaction	(1 − *α*)^−1^ − 1
C1.5	38	Chemical reaction	(1− *α*)^−1/2^

**Table 4 materials-14-01493-t004:** The characteristic temperatures of coal series kaolinite (CSK) calcination at different heating rates.

Heating Rate (°C/min)	Temperature (°C), Stage-I			Temperature (°C), Stage-II		
	Starting Temperature	Peak Temperature	Ending Temperature	Starting Temperature	Peak Temperature	Ending Temperature
5	334.5	505.9	598.9	603.2	658.1	802.2
10	341.2	520.5	615.3	629.2	668.4	826.3
15	350.6	527.8	633.0	633.1	678.5	837.3
20	365.3	541.0	643.1	647.2	694.4	846.6

**Table 5 materials-14-01493-t005:** Kinetics results of CSK dehydration.

Method	Heating Rate (°C/min)	Activation Energy *E_a_* (kJ·mol^−1^)	Pre-Exponential Factor log *A* (s^−1^)	Linear Correlation Coefficient *r*	Sequence Number of Function
The general integration	5	188.32	9.82185	0.987462	17
	10	179.81	9.24424	0.987345	17
	15	170.41	8.58677	0.989268	17
	20	178.20	9.05905	0.988422	17
Mac Callum–Tanner	5	194.34	10.30910	0.989254	17
	10	186.10	9.74108	0.989288	17
	15	176.88	9.08622	0.991059	17
	20	184.91	9.58110	0.990282	17
Satava–Sestak	5	191.68	10.05920	0.989254	17
	10	183.89	9.53854	0.989288	17
	15	175.18	8.93795	0.991059	17
	20	182.77	9.38547	0.990282	17
Average		182.71	9.44588	0.989355	

**Table 6 materials-14-01493-t006:** Kinetics results for the first step of the decarburization of CSK.

Method	Heating Rate (°C/min)	Activation Energy *E_a_* (kJ·mol^−1^)	Pre-Exponential Factor log *A* (s^−1^)	Linear Correlation Coefficient *r*	Sequence Number of Function
The general integration	5	160.36	9.00977	0.995881	17
	10	155.24	8.61711	0.995719	17
	15	153.77	8.49211	0.996107	17
	20	162.34	9.04798	0.995623	17
Mac Callum–Tanner	5	164.94	9.37296	0.996463	17
	10	160.12	8.99238	0.996383	17
	15	158.86	8.88716	0.996754	17
	20	167.67	9.46579	0.996290	17
Satava–Sestak	5	163.94	9.29719	0.996463	17
	10	159.36	8.95256	0.996383	17
	15	158.17	8.84942	0.996754	17
	20	166.49	9.37322	0.996290	17
Average		160.94	9.02980	0.996259	

**Table 7 materials-14-01493-t007:** Kinetics results for the second step of the decarburization of CSK.

Method	Heating Rate (°C/min)	Activation Energy *E_a_* (kJ·mol^−1^)	Pre-Exponential Factor log *A* (s^−1^)	Linear Correlation Coefficient *r*	Sequence Number of Function
The general integration	5	210.63	9.00166	0.995259	29
	10	211.22	9.05133	0.998382	29
	15	204.71	8.72449	0.994881	29
	20	217.44	9.35466	0.996899	29
Mac Callum–Tanner	5	219.45	9.68081	0.995962	29
	10	220.41	9.75185	0.996033	29
	15	214.06	9.42612	0.995686	29
	20	227.10	10.0888	0.997376	29
Satava–Sestak	5	211.38	9.29337	0.995962	29
	10	216.30	9.35931	0.996033	29
	15	210.30	9.06746	0.995686	29
	20	222.60	9.66119	0.997376	29
Average		215.47	9.37175	0.996295	

**Table 8 materials-14-01493-t008:** Mechanism equations and physical meaning of No. 17, No. 29, and No. 37.

Symbol of *g*(*α*)	Sequence Number of Function	Equation Name	Expression of *g*(*α*) Function
A2/3	17	Random nucleation and growth (*n* = 3/2)	[−ln(1 − *α*)]3/2
B3	29	Shrinking core (spherically symmetric)	1 − (1 − *α*)1/3
C2	37	Chemical reaction	(1 − *α*)−1 − 1

**Table 9 materials-14-01493-t009:** Kinetics results for the first mass loss stage of CSK calcination by TG.

Method	Heating Rate (°C/min)	Activation Energy *E_a_* (kJ·mol^−1^)	Pre-Exponential Factor log *A* (s^−1^)	Linear Correlation Coefficient *r*	Sequence Number of Function
Agrawal	5	147.19	7.18	0.998738	17
	10	163.56	8.32	0.996484	17
	15	147.22	7.22	0.998580	17
	20	147.65	7.24	0.997282	17
Satava–Sestak	5	152.40	7.61	0.998981	17
	10	168.30	8.68	0.997072	17
	15	152.96	7.69	0.998877	17
	20	153.56	7.71	0.997831	17
Average		154.11	7.71	0.997981	

**Table 10 materials-14-01493-t010:** Kinetics results for the second mass loss stage of CSK calcination by TG.

Method	Heating Rate (°C/min)	Activation Energy *E_a_* (kJ·mol^−1^)	Pre-Exponential Factor log *A* (s^−1^)	Linear Correlation Coefficient *r*	Sequence Number of Function
Agrawal	5	257.46	11.81	0.998230	37
	10	254.14	11.70	0.998209	37
	15	262.60	12.19	0.995548	37
	20	254.31	11.64	0.997943	37
Satava–Sestak	5	261.61	12.16	0.999009	37
	10	256.83	11.85	0.998378	37
	15	265.09	12.32	0.996072	37
	20	223.76	12.63	0.993748	37
Average		254.48	12.04	0.997142	

**Table 11 materials-14-01493-t011:** Comparison of kinetic results between TG and TG-FTIR-MS method.

Kinetic Method	Reaction	Activation Energy *E_a_* (kJ·mol^−1^)	Pre-Exponential Factor log *A* (s^−1^)	Linear Correlation Coefficient *r*	Sequence Number of Function
TG-FTIR-MS	dehydroxylation	182.71	9.44	0.989355	17
	the first step decarburization	160.94	9.02	0.996259	17
	the second step decarburization	215.47	9.37	0.996295	29
Traditional method (TG)	the mass loss of first stage	154.11	7.71	0.998000	17
	the mass loss of second stage	254.48	12.04	0.997123	37

## Data Availability

All data, models, and code generated or used during the study appear in the submitted article.

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
