# Peer review of "Kinetics of Dehydroxylation and Decarburization of Coal Series Kaolinite during Calcination: A Novel Kinetic Method Based on Gaseous Products"

_materials, 2021, doi:10.3390/ma14061493_

Round 1

Reviewer 1 Report

Dear authors,

The study discusses the kinetics of dehydroxylation and decarburization of carbon series kaolinite in the process of calcination. The authors propose an original and informative kinetic method based on analyzing the gaseous products. The manuscript is well written and structured. We have a few comments:

  1. Page 1, line 10. Please replace “decarbonization” with “decarburization”.
  2. Based on the XRD results, the Pb(Zr0.5Ti0.5)O3 phase was identified (Figure 1). However, Table 1, which presents the feed chemistry, does not show the content of lead and zirconium oxides.
  3. Please check Equation (8) on page 4.
  4. Table 2, page 3. Please replace “nitrogen” with “Nitrogen”.
  5. Table 3, page 3. Please replace the function g(α) with G(α).
  6. Table 3, page 4. The caption “Linear correlation coefficient r" is not readable.
  7. Throughout the text please replace oC with oC.
  8. Page 10, line 237. Table 5. Please replace “kinetics” with “Kinetics”.
  9. Table 5, page 10. The caption “Linear correlation coefficient r" is not readable.
  10. Page 11, line 238. Table 6. Please replace “kinetics” with “Kinetics”.
  11. Table 6, page 11. The caption “Linear correlation coefficient r" is not readable.
  12. Page 11, line 239. Table 7. Please replace “kinetics” with “Kinetics”.
  13. Table 7, page 11. The caption “Linear correlation coefficient r" is not readable.
  14. Table 10, page 12. The caption “Linear correlation coefficient r" is not readable.
  15. Page 13. 4. Conclusions. We suggest present the main results in more detail and more specifically, in order to demonstrate the novelty thereof.

Author Response

I have completed all revisions, please review. Thank you!

Reviewer 2 Report

Title:  should say "based" instead of "base".

Lines 18-19:  Assessing TG-FTIR-MS effectiveness is not within the scope of the paper. Furthermore, it was not defined what the authors mean by "effectiveness" in the paper. The authors should consider other terms when describing the proposed suitability of this analytical method.

Lines 24-25: The authors should reflect how applicable this is globally. I.e. which countries also have significant reserves/demand for CSK.

Line 29: Here, example industrial applications should be defined and referred to.

Line 62: More information should be provided with respect to how the sample was mined, at what depths, etc.

Section 2.1.1 XRD Analysis: More parameters of analytical methodology should be provided. Also, were there significant variations observed within the sample CSK?

Figure 1: Is this XRD spectrum an average of several measurements? How large were the variations?

Section 2.1.2 XRF Analysis: Also as above - more details on methodology. Standard deviations?

Section 2.1.3 Flammability analysis: Again, inter-sample variation.

Section 2.2 TG-FTIR-MS analysis: Why such gas composition was chosen? More details necessary on MS methodology.

Figure 2: It should say "2" instead of "1" in the legend. Also, was validation done to make sure that there is no significant variation across the sample? How many repeat runs were run?

Figure 3: Y axis title should say "Mass loss". Also, what are the experimental errors?

Figure 4: Peaks should be identified in terms of corresponding species here, too, so that the figure provides more information.

Figure 5: Is this also at 10 C/min?

Figure 10: This figure should indicate that the calculations are based on M44.

Tables 5, 6, 7, 10: Column titles are not all clearly visible.

Line 278: "efficient" should not be used here as it was not defined or studied. Another term could be used to highlight the proposed suitability/advantages of this analytical method.

Discussion: the authors should be more explicit when explaining what real world implications the better understanding of true calcination kinetics, and thus the value of the research, this paper offers.

Author Response

(The authors gave the same response as above.)
